# The Cytoprotective, Cytotoxic and Nonprotective Functional Forms of Autophagy Induced by Microtubule Poisons in Tumor Cells—Implications for Autophagy Modulation as a Therapeutic Strategy

**DOI:** 10.3390/biomedicines10071632

**Published:** 2022-07-07

**Authors:** Jingwen Xu, Ahmed M. Elshazly, David A. Gewirtz

**Affiliations:** 1School of Pharmacy, Guangdong Pharmaceutical University, Guangzhou 510006, China; jingwen_xu@gdpu.edu.cn; 2Massey Cancer Center, Department of Pharmacology and Toxicology, Virginia Commonwealth University, 401 College St., Richmond, VA 23298, USA; elshazlyam@vcu.edu; 3Department of Pharmacology and Toxicology, Faculty of Pharmacy, Kafrelsheikh University, Kafrelsheikh 33516, Egypt

**Keywords:** autophagy, microtubule poison, cytoprotective, cytotoxic, nonprotective, chemosensitization

## Abstract

Microtubule poisons, as is the case with other antitumor drugs, routinely promote autophagy in tumor cells. However, the nature and function of the autophagy, in terms of whether it is cytoprotective, cytotoxic or nonprotective, cannot be predicted; this likely depends on both the type of drug studied as well as the tumor cell under investigation. In this article, we explore the literature relating to the spectrum of microtubule poisons and the nature of the autophagy induced. We further speculate as to whether autophagy inhibition could be a practical strategy for improving the response to cancer therapy involving these drugs that have microtubule function as a primary target.

## 1. Introduction

This manuscript is the third in a series of papers that explores the role of autophagy in the response to therapeutic modalities in tumor cells. Our previous publications covered radiation [1] and cisplatin [2], and a paper is in progress relating to hormonal therapies in ER-positive breast cancer.

## 2. Autophagy and Cancer

Autophagy is a highly regulated catabolic process that plays a vital role in the maintenance of cellular homeostasis via the degradation of damaged organelles and cellular debris [3,4,5]. Autophagy occurs at a basal level in all cells and can be triggered by a variety of cellular signals and stresses including hypoxia, starvation, oxidative stress, endoplasmic reticulum (ER) stress and protein aggregation [6,7]. The autophagic process, which is tightly regulated by autophagy-related genes (*ATG*), is divided into five main steps: initiation, autophagosome nucleation, autophagosome formation, autophagosome-lysosome fusion, and cargo degradation [4].

Stress-based signals target the Unc-51-like kinase 1 (ULK1) complex to initiate autophagy, which in turn triggers phagophore nucleation by phosphorylation of the components of class III PI3K (PI3KC3) complex I [7,8,9]. PI3KC3 triggers phosphatidylinositol-3-phosphate (PI3P) production at the omegasome, [10,11] an ER-related structure. PI3P binds with its related domain on the omegasome and promotes the liberation of tryptophan–aspartic acid repeat (also known as WD40 or WD repeat) domain phosphoinositide-interacting protein 2 (WIPI2) as well as zinc-finger FYVE domain-containing protein 1 (DFCP1) [7,12,13]. WIPI2 binds ATG16L1, causing ATG12~ATG5–ATG16L1 complex recruitment, which promotes the ATG3-mediated conjugation of ATG8 proteins to phosphatidylethanolamine in the membrane [7,12]. Various cellular membranes, including the plasma membrane, recycling endosomes, mitochondria, as well as the Golgi apparatus, appear to be involved in autophagosomal membrane generation. The autophagosomal membrane is sealed, giving rise to a double-layered autophagosome, which fuses with the lysosome. The degradation of autophagic cargo is mediated by lysosomal acidic hydrolases, and the salvaged nutrients are returned back to the cytoplasm to be reused [7,14,15].

Several different functional forms of autophagy have been identified in cancer progression and promotion [16], specifically cytoprotective, non-cytoprotective, cytostatic and cytotoxic autophagy. *Cytoprotective* autophagy is a survival response that enables the tumor cells to buffer against starvation and evade apoptotic signals [17]. In many cases, the induction of protective autophagy, which decreases sensitivity to chemotherapeutic drugs and radiation, is associated with drug resistance [18]. Therefore, targeting the cytoprotective form of autophagy is considered as a potential therapeutic strategy in cancer treatment utilizing clinically approved autophagy inhibitors such as hydroxychloroquine (HCQ).

Autophagy induction may also contribute to tumor cell killing, either of its own, or by serving as a precursor to apoptosis, the form known as *cytotoxic* autophagy [16,19].

The *cytostatic* form of autophagy represses tumor cell growth independent from apoptosis. This form is most likely associated with the well-characterized and prolonged growth arrest condition known as senescence, which may be also contribute to tumor delay and dormancy [16,20].

A less well-appreciated function of autophagy is what we have termed the *nonprotective* form induced by chemotherapeutic agents or radiation whose suppression does not affect cell proliferation and apoptosis. As an example, in non-small cell lung cancer cells, autophagy inhibition via pharmacological and genetic interventions did not alter the sensitivity of p53 wild-type H460 cells to cisplatin [21]. Similarly, autophagy inhibition did not alter sensitivity to radiation of 4T1 breast tumor cells in cell culture, nor did chloroquine (CQ) alter the sensitivity of this cell line to radiation in an immune-competent animal model [22]. Furthermore, when non-protective autophagy is inhibited, there is no elevation in apoptotic cell death (unlike the case when cytoprotective autophagy is inhibited); however, the significance of this form of autophagy remains to be determined, since it is unclear of what advantage it might be to the cell under conditions of stress [22,23].

In this review, we explore the various forms of autophagy induced by microtubule poisons and their relevance to drug action and potentially to drug resistance.

## 3. Microtubules and Microtubule Poisons

Microtubules are composed primarily of α- and β-tubulin subunits, each of which have a GTP-binding site. The GTP-binding site on the α-subunit is a non-exchangeable site (N site) in that the bound GTP cannot be hydrolyzed or replaced by GDP. In contrast, the GTP (exchangeable E) binding site on the β subunit can be hydrolyzed to GDP [24,25]. The α- and β-heterodimers assemble into linear “protofilaments” that further assemble into a regular helical lattice around a hollow core [26]. The formed microtubules have plus and minus ends, and the process is dynamic. The dynamics of tubulin addition and release are faster for β-tubulin subunits exposed at the plus end of microtubules, while slower for α-tubulin subunits exposed at the minus end. Adding tubulin heterodimers to microtubules activates the GTPase activity of β-tubulin, which locks β-tubulin in the microtubule into a GDP-bound state [27].

Microtubules are central components of cell division and produce spindle bodies to form new daughter cells in late cell division. In addition to α- and β-tubulin, γ-, δ-, ε-, ζ- and η-tubulin are also found in eukaryotes [28]. Among these, γ-tubulin is found primarily around the centrioles, promoting intracellular microtubule nucleation and controlling mitotic spindle replication. ε- and δ-tubulin are newly discovered members of the tubulin superfamily, which maintain the microtubule cytoskeleton structure. In addition to these functions, microtubules are one of the major components of the cytoskeleton, supporting and maintaining basic cellular morphology, forming cilia and flagella, assisting cell motility, and serving as “tracks” for intracellular nutrient transport [28,29].

The first natural products belonging to the category of microtubule poisons with potent antitumor properties for clinical use were the vinca alkaloids, vinblastine and vincristine [30], identified in 1960. Approximately 10 years later, paclitaxel was isolated from *Taxus brevifolia*. Taxol was not approved by the FDA until 30 years after it was initially isolated and identified; only three years later, the more sensitive formulation of docetaxel was developed [31,32]. These drugs are classified into two main categories, depending on whether they act as microtubule destabilizing agents (the vinca alkaloids) or microtubule stabilizing agents (the taxanes). Cell proliferation, migration, invasion, and material transport are all dependent on dynamic changes in microtubule polymerization and depolymerization [27,33,34]. Thus, microtubule poisons continue to play a central role in the clinical treatment of both solid tumors or hematologic malignancies [35]. However, tumor cells can develop resistance by a number of mechanisms such as altered microtubule binding and efflux via the multidrug resistance pump family of transporters [36,37]. In addition, as described in detail in subsequent sections, resistance can develop via the promotion of the cytoprotective form of autophagy.

## 4. Direct Involvement of Microtubules in Autophagy

When microtubule-associated protein 1A/1B-light chain 3 (MAP1LC3, LC3) was identified as a key factor in the induction of autophagy, the Yoshimori laboratory had proposed the idea that microtubules might be involved in the progression of autophagy by affecting the efficient transport of autophagosomes to mammalian cells [38]. It was later confirmed that microtubules participate in the fusion of autophagosomes with lysosomes and in the formation of late autophagosomes [39,40].

The importance of microtubules in autophagic flux has long been recognized, and their role in autophagy initiation, trafficking, and lysosomal fusion has been continuously revealed in the last two decades. In addition to LC3, other autophagy-related proteins such as ULK1, Beclin-1, WIP1, ATG5, and ATG12 that are involved in autophagosome formation were found to be associated with microtubules as well [41,42,43]. Dynamic changes in microtubule and post-translational modifications of microtubule play an important role in regulating starvation-induced autophagy, as microtubule acetylation occurs prior to autophagosome formation in response to nutrient deficiency. Acetylation modification signals kinesin recruitment to microtubules, followed by JNK activation and Beclin-1 release from the Beclin-1-Bcl-2 complex to initiate autophagy [41].

AMBRA1 is a key factor in the regulation of autophagy in vertebrates. AMBRA1 promotes the interaction of Beclin-1 with its target lipid kinase, VPS34, which mediates autophagosome nucleation [44]. Moreover, AMBRA1 serves as a direct regulatory link between ULK1 and Beclin-1-VPS34, which is required for the localization and activity of the intracellular core complex. Once the autophagosome is formed, it moves along the microtubule in both directions (minus-end and plus-end) via kinesin motor complexes, accumulating at the microtubule-organizing center, and eventually moving towards the lysosome [39,40,45]. Their centripetal movement is dependent on the motor protein dynein and is important for their fusion with lysosomes. In addition to the function of microtubules mediating autophagosome transport, there is a strict regulatory relationship between cytoskeletal dynamics and autophagosome formation [43].

The effect of microtubules on the fusion of autophagosomes with lysosomes is controversial. One view is that microtubule dynamics does not affect the fusion of autophagosomes with lysosomes and that this fusion can occur in the presence of microtubule poisons [46,47]. However, by combining real-time observation and microinjection techniques, other investigators proposed that after formation, autophagosomes utilize a dynein-microtubule system to rapidly move toward lysosomes located near the centrosome [40]. When cells are starved for glucose, the cyclin-dependent kinase inhibitor p27Kip1 (p27) promotes autophagy by maintaining elevated microtubule acetylation via an ATAT1-dependent mechanism to promote autophagosome transport along microtubule trails [48].

In addition to their classification into the two main categories, namely microtubule destabilizing agents and microtubule stabilizing agents, microtubule poisons are then subdivided into the following categories according to their specific binding domains to microtubule proteins (Figure 1): colchicine site, vinca alkaloid site, taxane site, laulimalide site and epothilone site. Analysis of autophagy induced by different microtubule poisons in tumor therapy is based on these subcategories.

### 4.1. Colchicine Site

*Colchicine.* Colchicine, an alkaloid isolated from plants belonging to the genus *Colchicum* (*Autumn crocus*), is a classical antimitotic compounds, but is not actually used in cancer therapeutics due to its high degree of toxicity [49]. Colchicine blocks mitotic cells in metaphase through the formation of poorly reversible tubulin–colchicine complexes. In addition, colchicine binds to microtubule ends, thereby preventing elongation [49].

Bhattacharya et al. [50] studied the effect of colchicine in the A549 non-small cell lung cancer cell line and reported that, although colchicine at a clinically relevant concentration of 2.5 nM showed no cytotoxic effect or cytostatic effects, it substantially inhibited microtubule formation. Colchicine treatment promoted senescence based on beta-galactosidase staining, cell flattening, increased expression of p53 and p21. Autophagy induction mediated by ROS was confirmed by MDC staining, acridine orange staining, and increased Beclin-1 and LC3B levels in A549 cells at this concentration. The combination of colchicine and the autophagy inhibitor 3-methyladenine (3-MA) attenuated senescence and caused apoptotic cell death. The selectivity of this combination was based on studies in the normal lung fibroblast cells (WI38). The authors suggested that autophagy in these cells had a cytoprotective effect under colchicine-induced stress [50]. However, it must be recognized that 3-MA is a relatively non-specific inhibitor of autophagy, and in the absence of other pharmacologic and genetic inhibitor studies, unequivocal conclusions as to the nature of the autophagy cannot be drawn.

Although colchicine has potent antitumor properties that are attributed to the irreversible binding with tubulin causing cell cycle inhibition and the induction of apoptosis, colchicine lacks selectivity against tumor cells. This limitation has led to the development of colchicine derivatives as well as colchicinoid prodrugs with less toxicity and more specific targeting to tumor cells [51].

An association between autophagy and colchicine derivatives has been reported in a number of studies. JG-03-14, a substituted pyrrole colchicine mimetic that binds to the colchicine site of tubulin, induced a significant reduction in the viability of MCF-7 and MDA-MB-231 cells [52]. Importantly, JG-03-14 was able to induce autophagy in up to 70% of the MCF-7 cell population by the third day of treatment with minimal apoptosis (as evidenced by the TUNEL assay). Conversely, JG-03-14 induced a significant amount of apoptosis by the third day of treatment in the MDA-MB231 cell line, but also demonstrated a significant degree of autophagy, based on acridine orange staining [52]. These findings suggest that JG-03-14 may have induced autophagic cell death in these breast tumor cell lines, although the precise function of the autophagy was not directly assessed in this work.

Studies of the antitumor action of JG-03-14 were further extended in B16/F10 melanoma and HCT-116 colon cancer cell lines. JG-03-14 treatment resulted in a significant reduction in colony forming capacity for both B16/F10 and HCT-116 cell lines, with a small fraction of the population undergoing apoptosis. There was also clear evidence of autophagic flux based on electron microscopy, acridine orange staining, RFP-LC3 redistribution as well as p62/SQSTM1 degradation in the case of the HCT-116 cells; the residual surviving cells entered senescence [53]. Autophagy inhibition by CQ showed only a moderate degree of protection from JG-03-14 cytotoxicity in the HCT-116 cells, suggesting that the cytotoxic function of autophagy might have played only a minor role in the action of JG-03-14. The B16/F10 melanoma cells showed a similar trend when treated with another autophagy inhibitor, bafilomycin, in that drug cytotoxicity was only minimally altered. Studies of JG-03-14 in B16/F10 murine tumors in female B6C3F1 showed no effect on the number of lung nodules developed, but a marked reduction in the nodules size [53]. However, there were no experiments involving the use of autophagy inhibitors in vivo. These studies were performed in the earlier days of the autophagy field, and before the advent of many of the more sophisticated genetic approaches that are now used routinely.

Larocque et al. reported that a novel derivative of allocolchicine, known as Green 1, reduced the viability of PANC-1 pancreatic cancer cells [54]. Moreover, Green 1 decreased the proliferation of E6-1 and Jurkat human T cell leukemia cells as well as disrupted cell membrane integrity, confirmed by the elevation of trypan blue-positive cells. Selectivity was demonstrated by the drug having a minor effect on normal human fibroblasts. Green 1 treated cells did not show apoptotic morphology, while autophagy characteristics were apparent, including LC3-I/LC3-II conversion as well as Beclin-1 expression. Here again, it is possible that Green 1 triggered autophagic cell death selectively in pancreatic cancer and acute T cell leukemia cell lines; however, this possibility was not directly assessed [54].

Fang et al. studied the effect of AD1, a colchicine derivative, in human malignant glioblastoma cells, observing a reduction in cell viability as well as time-dependent cell death in both U87MG and U373MG cell lines. Autophagy was observed by LC3I/LC3II conversion and acridine orange staining, together with flow cytometry. Although AD1-induced autophagy could be a critical factor in triggering glioma cell death, this was also not directly assessed in this work [55].

*Podophyllotoxin*. Podophyllotoxins (PTOX) are a subclass of plant secondary metabolites known as lignans that present in the resin of Podophyllum plants. PTOX antitumor properties are attributed to both the ability to disrupt microtubules and the inhibition of topoisomerase II [56,57]. PTOX inhibits the assembly of microtubules by targeting the colchicine binding site, which results in the inhibition of the polymerization of microtubule proteins as well as causing G2/M cell cycle arrest [58]. In addition, some PTOX derivatives, such as etoposide and teniposide, have the ability to stabilize the cleavage complex between topoisomerase II and its DNA substrate, which results in DNA breakage, ultimately resulting in tumor cell death via the accumulation of chromosomal damage [56,59].

Only a limited number of studies have investigated the possible relationship(s) between autophagy and PTOX (our focus being primarily on microtubule poisons) and its derivatives. Choi et al. [60] highlighted the possible involvement of autophagy in triggering cellular death by podophyllotoxins. In studies of the effect of one PTOX derivative, PTOX acetate (PA), in A549 and NCI-H1299 human non-small cell lung cancer cell lines, PA disrupted microtubule polymerization and increased cell cycle arrest, as well as triggered apoptotic cellular death in a time-dependent manner. They further reported that PA induces a time-dependent autophagy in A549 and NCI-H1299 cell lines as well as increases the expression of the autophagic markers ATG3, ATG5, ATG7, and Beclin-1 and promoting LC3 cleavage, raising the possible cytotoxic role of PA-induced autophagy [60].

Wang J. et al. [61] synthesized PTOX indirubin hybrids and investigated their effects against two human chronic myeloid leukemia K562/VCR cell lines. They reported that these compounds promoted apoptosis and cell cycle arrest in K562/VCR cell lines to various degrees with the most potent derivative being Da-1, which showed strong antiproliferative activity against the resistant K562/VCR cell lines. Importantly, they reported that Da-1 could induce autophagy in the resistant K562/VCR cells by increasing the expression levels of Beclin1 and LC3-II by Western blotting [61]. It was suggested that DA-1 mediates its antiproliferative activities, in part, by inducing cytotoxic autophagy.

Ren et al. [62] studied the effect of a novel PTOX derivative, OAMDP (4β-(1,3,4-oxadiazole-2-amino-5-methyl)-4-deoxypodophyllotoxin), in the HepG2 hepatocarcinoma cell line. OAMDP demonstrated strong inhibitory effects on HepG2 cell survival and proliferation in time and dose-dependent manner, with significant promotion of apoptosis. They reported that OAMDP also induces autophagy in HepG2 cells, as indicated by Monodansylcadaverine (MDC) staining and LC3 conversion; although suggestive of a cytotoxic autophagy induced by OAMDP, this possibility was never directly examined [62].

As was the case with drugs targeting the colchicine binding site, a definitive role for PTOX-induced autophagy in mediating drug action remains uncertain in the absence of rigorous experiments involving genetic and pharmacologic inhibition of autophagy.

*Combretastatin A-4*. Combretastatins represent a large family of bioactive stilbenes, dihydrostilbenes, phenanthrenes and macrocyclic lactones named Combretastatins A, B, C, and D, respectively [63]. Combretastatins are isolated from *Combretum caffrumtree* and show favorable anticancer activities. A well-studied member of this drug family is Combretastatin A-4 (CA-4), a tubulin-depolymerizing agent that binds at the colchicine binding site on the β-tubulin subunit of tubulin, resulting in depolymerization and destabilization of tubulin polymers of the cytoskeleton, causing an increase in vasculature permeability and disruption of the tumor blood flow [63,64,65,66].

Li et al. investigated the potential relationship between CA-4 and autophagy in various cancer cell lines including MDA-MB-231 breast tumor cells, SGC-7901 human gastric tumor cells, and SMMC-7721 human hepatocellular carcinoma cells. These investigators identified CA-4 induced autophagy in these cell lines, which was confirmed by p62/SQSTM1 degradation, acridine orange staining of intracellular acidic vesicles, LC3-I to LC3-II conversion as well as GFP-LC3 punctate fluorescence. They reported that pretreatment with the pharmacologic autophagy inhibitors, 3-MA and bafilomycin A1, or siRNAs against Atg5 and Beclin 1 genes potentiated the apoptotic cell death mediated by CA-4 in SGC-7901 and SMMC-7721 cell lines, providing relatively unequivocal evidence for the cytoprotective function of CA-4 mediated autophagy in these cell lines [67].

The cytoprotective function of autophagy was confirmed utilizing CA-4 and 3-MA in vivo with the SGC-7901 xenograft tumor model. Although the utilization of 3-MA in tumor-bearing animals is relatively rare in the scientific literature, the combination of CA-4 and 3-MA exerted significant antitumor effects against SGC-7901 xenografts compared with each drug alone. The increased antitumor activity was confirmed by evidence for cleavage of -caspase-3 and PARP, markers of apoptosis, in extracts of the tumor xenografts. Importantly, the decline in the expression of LC3-II tended to confirm that the autophagy was inhibited in the tumor bearing animal models [67].

Additional studies were performed to investigate potential pathways involved in CA-4-mediated autophagy. Here, it was shown that a JNK inhibitor or JNK siRNA suppressed autophagy, as indicated by the inhibition of CA-4-induced LC3-II production and reduced Bcl-2 phosphorylation accompanied by increased CA-4-mediated apoptosis in the SGC-7901 cells [67]. In addition, pretreatment with the Bcl-2 inhibitor, ABT-737, resulted in autophagy inhibition by blocking LC3-I conversion to LC3-II and significantly augmenting CA-4 anticancer activity in the SGC-7901 cells. These experiments suggested that the JNK-Bcl-2 pathway may play a critical role in the CA-4-mediated protective autophagy [67].

Studies by Greene et al. [68] supports the premise that the role of autophagy is not only cancer type dependent but is also dependent on the cell line. These investigators showed that different combretastatins, including CA-4 and CA-432, induce a significant reduction in the viability of adenocarcinoma-derived cell lines. They reported that combretastatin-induced cell death in the CT-26 cell line was not suppressed by 3-MA or BAF-A1 mediated autophagy inhibition, and furthermore was not augmented by rapamycin-mediated autophagy activation, indicating the autophagy in CT-26 cell line is likely to be *non-protective*. In contrast, inhibition of the autophagic pathway in HT-29 cells enhanced the activity of CA-432, suggesting that autophagy may have a role in facilitating the survival of HT-29 cells, i.e., that here the autophagy was *cytoprotective* [68].

Wang et al. [69] also demonstrated that CA-4 could induce cell-protective autophagy as evidenced by a marked increase in the LC3-I to LC3-II conversion and autophagosome accumulation in SJSA and MG63 human osteosarcoma (OS) cell lines, and that the treatment with CA-4 and the autophagy inhibitor CQ resulted in a synergistic cytotoxic effect in the OS cell lines [69]. These results further support the possible cytoprotective role of CA-4- induced autophagy, which will require further confirmation with genetic inhibition of autophagy related genes.

Combretastatin A4 phosphate is a prodrug for CA-4 that has shown a potential anti-cancer effect in Phase I clinical trials [70]. Hoang et al. [71] showed that the anti-tumor effect of CA-4 phosphate is enhanced in autophagy-defective PC3 prostate cancer xenografts (developed with retrovirally transducing PC-3 cells with ATG4BC74A, an inactive and dominant-negative mutant of the autophagy related gene atg4B) compared with controls. Significant central necrosis as well as a higher number of senescent cells were evident in autophagy-defective PC3 xenografts both 24 h and 1 week following CA4P treatment, indicating the possible role of autophagy inhibition (i.e., cytoprotective autophagy) in enhancing the antitumor effects of CA4 phosphate via lowering the threshold of peripheral tumor cells to tolerate the CA4 phosphate-induced metabolic stress [71,72].

Taken together, these findings suggest that combretastatin-mediated autophagy is largely cytoprotective but can also be dependent on both the cancer type as well as the specific cell line studied.

### 4.2. Vinca Alkaloid Site

Vinca alkaloids are a class of organic compounds that were isolated from the leaves of the Madagascar periwinkle plant, *Catharanthus roseus*. Five distinct vinca compounds with significant antineoplastic activities have been identified, specifically vinblastine, vincristine, vindesine, vinorelbine, and vinflunine. Vinblastine and vincristine continue to be two of the most commonly used anticancer agents. These drugs are structurally related except that vincristine contains an aldehydic functional group attached to the nitrogen of the indole moiety whereas vinblastine contains a methyl group. This minor difference in structure results in significant differences in both the antineoplastic activities and the toxicity between the two agents [73]. Vinblastine has been utilized in the clinical treatment of leukemia, non-Hodgkin’s and Hodgkin’s disease, breast cancer, testicular carcinoma, and small-cell lung cancer. Vincristine have been used for many years in the treatment of malignancies including acute lymphoblastic leukemia, B-cell lymphoma, metastatic melanoma, and Wilms’ tumor [74].

Vinca alkaloids bind the vinca binding site on microtubules causing microtubule interruption and dissociation [75]. Vinca alkaloids tend to demonstrate cell cycle specificity for the M-phase [76]. At low concentrations, these drugs decrease the rates of both growth and shortening at the microtubule assembly end, blocking mitosis, with the cells eventually undergoing apoptosis [77]. At high concentrations, vinca alkaloids cause microtubule depolymerization and the destruction of mitotic spindles. The dividing tumor cells show condensed chromosomes and appear to be blocked in mitosis [78]. Furthermore, vinca alkaloids exhibit antiangiogenic and antivascular activities, inducing potent vascular disruption, and ultimately leading to tumor necrosis [36,79].

*Vinblastine.* The relationship between vinblastine and autophagy was mentioned earlier in a study made by Punnonen et al. [80] on Ehrlich ascites tumor cells where vinblastine was shown to potently induce autophagic vacuole accumulation and to interrupt autophagic flux [80]. Recent studies demonstrated that vinblastine impairs the cytoskeleton-mediated transport of vesicles, preventing autophagosome movement and their fusion with lysosomes, which results in a blockade to autophagosome maturation/degradation and, thereby, autophagic flux [47,81]. Vinblastine’s ability to perturb microtubule function as well as the inhibition of autophagic flux, may account for its antineoplastic potency. The potential relationship between vinblastine-induced autophagic inhibition and its side effects will require further investigation.

Adiseshaiah et al. [82] studied vinblastine (autophagosome maturation/degradation blocker [81]) in combination with C6-Ceramide (autophagy inducer [82,83]) in human hepatocarcinoma (HepG2) and colon cancer (LS174T) cell lines. They reported a synergetic elevation of apoptotic cell death over that of each drug alone with a significant accumulation of autophagic vacuoles and blocked autophagy maturation, evidenced by LC3-II protein elevation together with p62/SQSTM1-protein accumulation. C6-Ceramide in combination with CQ also showed a synergistic effect over that of C6-ceramide alone.

The combination of C6-Ceramide and vinblastine was also tested in vivo in female nude mice implanted subcutaneously with LS174T human colon cancer cells. This combination resulted in a statistically significant suppression in tumor growth exceeding that of each drug alone [82], suggesting that autophagy may play a cytoprotective role; however, this conclusion awaits further studies.

*Vincristine.* Belounis et al. [84] studied autophagy in different neuroblastoma (NB) cell lines and reported that vincristine (VCR) induced autophagy in a dose dependent manner in vitro. These data were confirmed by an in vivo study that showed autophagy activation in NB tumors from mice treated with VCR compared to non-treated tumors. Moreover, they showed that *ATG5*-silenced cells were significantly more sensitive to VCR than the controls, indicating the possible protective role of VCR-induced autophagy. Furthermore, mice treated with the combination of HCQ and VCR developed significantly fewer extended tumors compared with mice treated with VCR alone. These data suggested that autophagy inhibition sensitizes NB cells to chemotherapy, which supports a cytoprotective function of autophagy in this experimental model [84].

Shan et al. [85] recently studied the effect of autophagy inhibition in a number of cell lines, including the VCR-resistant esophageal cancer cell line Eca-109/VCR. A new series of 5-amino-2-ether-benzamide derivatives were synthesized as potential autophagy inhibitors. The most potent of these, 4d (N-(cyclohexylmethyl)-5-(((cyclohexylmethyl) amino) methyl)-2((4(trifluoromethyl) benzyl) oxy) benzamide) in combination with VCR demonstrated a significant synergistic anti-proliferative effect against Eca-109/VCR cells as compared to the cells treated with each drug alone [85], supporting the cytoprotective role of VCR-induced autophagy. In agreement with these findings, Sheikh-Zeineddini et al. [86] reported that the treatment of both pre-B acute lymphoblastic leukemia (ALL) REH and Nalm-6 cell lines with CQ causes a reduction in cell viability either as a single agent or in combination with VCR, indicating that autophagy suppression in pre-B ALL cells could sensitize tumor cells to these microtubule poisons [86].

Sun et al. [87] studied the Glycosylphosphatidylinositol-anchored protein, CD24, which is overexpressed in human retinoblastoma cells and is associated with retinoblastoma cell sensitivity in response to VCR therapy. Autophagy inhibition with CQ was shown to sensitize retinoblastoma cells to vincristine therapy; these studies are limited, however, in the absence of experiments involving genetic autophagy inhibition, as is much of the work in this field.

CD24 knockdown was further demonstrated to markedly decrease autophagic flux, inhibit the formation of autophagosomes and increase the sensitivity of retinoblastoma cells to vincristine therapy. CD24 downregulation caused PTEN depletion and increased the phosphorylation levels of PTEN, AKT, and mTOR. PTEN depletion significantly suppressed autophagy and sensitized retinoblastoma cells to vincristine therapy. [87]. These data indicated that CD24 triggers the cytoprotective form of autophagy via the PTEN/AKT/mTORC1 signaling pathway, and consequently decreases the sensitivity of retinoblastoma cells to VCR.

Li et al. [88] studied the effects of Matrine (MAT) in the K562/ADM cell line. MAT is an alkaloid that sensitizes K562/ADM cells to Adriamycin and vincristine, and at the same time, promotes autophagy, as evidenced by autophagic vacuole accumulation, increased LC3 punctate fluorescence, increased LC3 II protein levels, and p62/SQSTM1 protein degradation. CQ significantly reversed the sensitization mediated by MAT on the viability of K562/ADM cells. It was therefore suggested that MAT reverses drug resistance to Adriamycin and vincristine via promoting a cytotoxic form of autophagy [88].

Consistent with this view from the studies by Li et al. [88] and Takahashi et al. [89] reported that cotylenin A, which is a plant-growth regulator, in combination with vincristine, synergistically inhibited the growth of myeloma cells and induced apoptosis. Interestingly Cotylenin A with vincristine synergistically induced autophagy, as evidenced by LC3-II accumulation and p62/SQSTM1 degradation. These results indicate that autophagy may play a cytotoxic role, although more direct studies would be required to support this conclusion [89].

Collectively, these data indicate that the function of autophagy is likely to vary between different experimental tumor models. In some cases, vincristine cytotoxicity may be hindered by the cytoprotective form of autophagy, while in other cases cytotoxic autophagy is expressed.

### 4.3. Taxane Site

*Paclitaxel.* Paclitaxel, an antimitotic agent originally extracted from the bark of the Pacific yew tree, was identified and developed by Dr. Susan Band Horwitz in 1979 [90]. Paclitaxel binds to the β-tubulin subunit and forms stable and nonfunctional microtubules, thereby blocking cancer cell growth by interrupting cell division at the metaphase/anaphase transition, resulting in cell death [91,92]. This is a mechanism quite different from that of vincristine and vinblastine, which cause the disassembly of microtubules. Since its approval by the FDA, Taxol, a semisynthetic form of paclitaxel, has expanded treatment options for patients with breast [93] and ovarian cancers [94]. Non-small cell lung cancer, pancreatic cancer, and AIDS-related Kaposi sarcoma are all sensitive to Taxol as well [95,96].

The autophagy caused by paclitaxel in different types of tumor cells appears to be drastically different. Zou et al. found that paclitaxel *cannot* induce autophagy in SKBr3 and MDA-MB-231 breast cancer cells, unless ARHI (DIRAS3) is re-expressed in the cells [97]. Veldhoen et al. found that low concentrations of paclitaxel inhibited autophagy. It was suggested that paclitaxel-induced mitotic arrest leads to decreased autophagic flux through phosphorylation and inhibition of Vps34 and subsequently results in aberrant autophagosome trafficking and localization, which in turn inhibits autophagosome degradation. By detecting autophagosome formation of GFP-LC3 fluorescence in single cells and cell death using flow cytometry, these investigators demonstrated that 3-MA, siRNA ATG7, or siRNA VPS34 reduced paclitaxel-induced apoptosis, suggesting that the blocked autophagy still plays a key role in paclitaxel-induced cell death [98]. These studies appear to support a direct role of (cytotoxic) autophagy in paclitaxel-induced cell death.

In contrast, Zhang et al. [99] found that paclitaxel increases autophagosome formation and autophagic flux in A2780, 3AO and SKOV3 ovarian cancer cells, supported by GFP-LC3 puncta upregulation that is increased after BafA1 treatment, as well as p62/SQSTM1 degradation. Similar to this study, paclitaxel not only promoted the formation of acidic vesicular organelles stained with acridine orange, but abrogation of Beclin-1 or use of the pharmacological autophagy inhibitor, 3-MA, enhanced the cytotoxicity of paclitaxel in A549, U87, PC3 and HT-29 cells, suggesting that paclitaxel induced cytoprotective autophagy [99,100].

This contradiction in the results of preclinical studies is also reflected in clinical trials. The combination of gemcitabine/nab-Paclitaxel with HCQ in pancreatic cancer did not improve the primary endpoint of overall survival at 12 months (NCT01506973) [101]. However, recently, a phase II study of the efficacy and safety of CQ in combination with Taxanes in the treatment of patients with advanced or metastatic anthracycline-refractory breast cancer appeared to be efficacious (NCT01446016) [102]. A similarly promising outcome was also reported in a clinical trial of HCQ for paclitaxel-resistant non-small cell lung cancer (NCT00728845, NCT01649947) [103]. From these inconsistent findings, it can be surmised that both the dose of paclitaxel and the type of tumor on which it acts are factors that affect autophagic flux and the nature and role of autophagy.

*Docetaxel.* Docetaxel is a tetracyclic diterpenoid and a secondary alpha-hydroxy ketone that is an analogue of paclitaxel. Therefore, these two drugs share common mechanisms in cancer therapy. However, in terms of autophagy, docetaxel seems to be less similar to paclitaxel, particularly in prostate cancer cells. Studies found that docetaxel induced autophagy in LNCaP, PC3 and DU145 prostate cancer cells [104,105], supported by increased LC3-II conversion and LC3 puncta, but that co-treatment with 3-MA, rapamycin (in non-cytotoxic concentration) and ATG5 siRNA do not alter docetaxel toxicity or resistance, showing a non-protective role [105]. In contrast, an earlier study showed that 3-MA altered docetaxel-induced toxicity in PC-3 cells [106]. However, the absence of more specific and rigorous autophagy inhibition experiments, such as silencing of autophagy-related genes, places this conclusion into question. The outcomes of clinical trials utilizing docetaxel and HCQ are also currently unknown due to the termination of the trial (NCT00786682).

### 4.4. Epothilone Site

*Epothilone A and epothilone B*. Epothilones are a type of natural cytotoxic compound belonging to 16-member natural macrolides. Thus far, six types of epothilone and derivatives have been reported. Epothilone A and B were first isolated from myxobacterium *Sorangium cellulosum*, and the efficiency of epothilone B was shown to be higher than that of epothilone A [107]. Similar to paclitaxel, epothilone also exerts its antitumor activity by stabilizing cell microtubules, but the binding sites of epothilone and paclitaxel are different. Among the five oxygen-containing polar groups constituting the macrocycle of epothilone, only C7-OH is located near the similar C7-OH part of paclitaxel, and the polymerization activity of epothilone B is 2 to 10 times higher than that of paclitaxel [108]. In addition, epothilone is less susceptible to multidrug resistance pump-mediated efflux compared to paclitaxel, and the expression of MDR is not altered in epothilone-resistant cell lines, implying a wider choice for chemo-resistant patients [109,110].

Kroemer’s laboratory [111] established a cell model in which cells selected for resistance to epothilone B can only survive in the continuous presence of 100 to 300 nM epothilone B (termed A549-B480 cells). They found that in the A549-B480 cells, autophagy was absent and that this absence was correlated with the lack of an essential autophagy-related protein, Atg7. In contrast, parental A549 cells treated with epothilone B manifested all signs of bona fide autophagy (GFP-LC3 puncta and the roundish structures observed by electron microscopy). Their findings identify a new epothilone B resistance-associated autophagic defect.

Similar to paclitaxel, epothilone B was shown to induce autophagy in tumor cells [111,112]. Upon co-treatment with different concentration of the pharmacologic autophagy inhibitor, bafilomycin A1, Rogalska et al. [113] reported that autophagy inhibition increased epothilone B sensitivity in SKOV-3 and OV-90 human ovarian cancer cells. However, no other autophagy inhibition strategies were used in this study. Despite researchers’ confidence in the antitumor effects of epothilone B, a clinical trial that included 799 patients showed that the median progression-free survival of epothilone was inferior to that of paclitaxel [114]. As we noted in Section 4.3, Veldhoen et al. [98] showed that paclitaxel-induced autophagy promotes breast cancer cell death, while cytoprotective autophagy induced by epothilone B may be responsible for the weakened sensitivity. However, there is a dearth of studies on the induction of autophagy in cancer cells by epothilone, and there are some limitations to the existing experimental designs, such as an insufficient number of strategies to inhibit autophagy or the detection of autophagic flux. These shortcomings are not sufficient to support the hypothesis that the poor clinical outcome of epothilone is solely attributable to cytoprotective autophagy, and more rigorous studies are needed.

*Ixabepilone*. Ixabepilone is an epothilone B analog, approved as monotherapy for locally advanced or metastatic breast cancer resistant or refractory to anthracyclines, taxanes, and capecitabine. Thus far, the role of autophagy induced by ixabepilone has not been consistently reported. Tanei et al. [115] reported in 2016 that ixabepilone increases p62/SQSTM1 levels but has no effect on the basal levels of LC3B-II in MDA-MB-231 and SUM159 cells; therefore, these data are insufficient to support the conclusion that ixabepilone upregulated autophagy. A more comprehensive later study showed that ixabepilone increased the expression of LC3-II, and that CQ enhanced this trend in MDA-MB-231 and BT 594 cells. It was also found that CQ, 3-MA and Beclin-1 siRNA increased ixabepilone sensitivity. Most importantly, combinatorial treatment with ixabepilone and CQ enhanced antitumor activity in a breast cancer nude mice model, suggesting that cytoprotective autophagy was induced by ixabepilone. [116]. One difference between these two studies relates to the concentration of ixabepilone used, with the latter having a minimum study concentration 10 times higher than the former. The latter study is consistent with the findings regarding the induction of autophagy in ovarian cancer cells by epothilone B [113].

### 4.5. Laulimalide/Peloruside Site

*Laulimalide and Peloruside A*. Laulimalide is a potent microtubule stabilizer that was originally isolated from the sponge *Cacospongia mycofijiensis* [117]. Peloruside A was isolated from the New Zealand marine sponge Mycale hentscheli in 2000 [118]. Researchers found that peloruside A and laulimalide compete for the same or overlapping binding sites, not taxane site but the M-loop of β1 and loop H1−B2 of β2 [119,120,121]. In contrast to paclitaxel, peloruside A and laulimalide are not substrates for the multidrug resistance P-glycoprotein efflux pump and are not affected of β-tubulin mutations in the taxane binding site [119]. These properties indicate that laulimalide and peloruside A have the potential to treat paclitaxel resistant tumors, but no clinical trials are currently in progress due to a limited drug supply, unstable drug structure [122] and the lack of antitumor activity studies in vivo.

## 5. Summary and Overview

Although many new cancer treatments have been developed in recent years, such as targeted therapy and immunotherapy, microtubule poisons remain a critical class of first-line chemotherapeutic agents. However, the relationship(s) between microtubule poisons and autophagy are exceedingly complex. This is likely due to a number of factors including that (a) microtubules play a direct role in the autophagic process, as described above; (b) the different microtubule poisons do not all act at the identical microtubule binding sites, as indicated in Figure 1; (c) autophagy cannot have only one (cytoprotective) function, but (at least) three others, termed cytotoxic, cytostatic and nonprotective. As indicated in Table 1, although the cytoprotective function is often clearly induced, both the cytotoxic and nonprotective function have also been identified in response to these agents. This may be dependent upon the chemical structure of the drug as well as the experimental cell line being utilized. A final and critical issue is that the literature evaluating the role of autophagy in response to microtubule poisons in tumor cells has generally relied on pharmacologic autophagy inhibitors such as CQ (or HCQ) and 3-MA drugs, which are not exclusively autophagy inhibitors [106]. In the absence of studies utilizing genetic autophagy inhibition, inferences related to the nature of the autophagy are not sufficiently well supported to be conclusive.

Currently, several clinical trials targeting autophagy inhibitors in combination with microtubule poisons are in progress combining the pharmacological autophagy inhibitors, CQ or HCQ, with taxanes for the treatment of solid tumors. In the pancreatic cancer study with nab-Paclitaxel, the authors concluded that “Hydroxychloroquine added to chemotherapy did not improve overall survival”. In the Phase II breast cancer study using chloroquine, the overall response rate was ~45% as compared to an expected overall response rate of 30%. In a Phase Ib/II study of hydroxychloroquine in combination with various agents, including paclitaxel, the authors concluded that “Addition of hydroxychloroquine is safe and tolerable with a modest improvement in clinical responses…”. The question is whether autophagy inhibition is likely to be incorporated into standard of care treatment strategies with microtubule poisons. We would argue that this is unlikely due to the fact that, as described in this paper, there is no consistent form of autophagy that is induced by the spectrum of agents in different tumor models. This issue is exacerbated by the fact that we have no clinically approved methodology for assessing autophagy induction in solid tumors or, furthermore, to distinguish between the functional forms that might be induced. Finally, the closely related question is that we cannot assess/monitor whether the autophagy inhibitor (in the current cases, chloroquine or hydroxychloroquine) is actually effective at suppressing autophagy in the tumor cells [101,102,103].

## Figures and Tables

**Figure 1 biomedicines-10-01632-f001:**
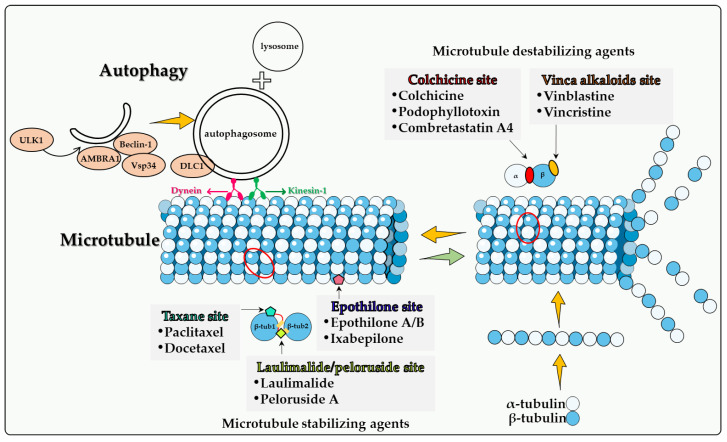
**The binding sites of microtubule poisons and autophagy.** Kinesin-1 and dynein are microtubule motors that transport intracellular cargos. Once formed, the autophagosome travels along the microtubule via kinesin motor complexes, accumulating at the microtubule-organizing center, and eventually moving towards the lysosome. This centripetal movement is dependent on the motor protein dynein and is required for their fusion with lysosomes. Microtubule poisons are classified into two main categories, depending on whether they act as microtubule destabilizing agents or microtubule stabilizing agents. Microtubule destabilizing agents: Colchicine, Podophyllotoxins and Combretastatin A-4 target the colchicine site between α- and β-tubulin subunits (red dot); Vinca alkaloids bind to the β-tubulin subunit of the α/β-tubulin heterodimer (yellow dot). Microtubule stabilizing agents: Paclitaxel and Docetaxel target the Taxane site, binding to the β-tubulin subunit (green pentagon). Laulimalide and Peloruside A binding involves the bridging of two adjacent tubulin dimers (β-tubulin1 and β-tubulin2) across protofilaments in microtubules (light green squares). Epothilone sites and Taxane sites are different. Among the five oxygen-containing polar groups constituting the macrocycle of epothilone, only C7-OH is located near the similar C7-OH moiety of paclitaxel, and the polymerization activity of epothilone B is 2 to 10 times higher than that of paclitaxel (pink pentagon).

**Table 1 biomedicines-10-01632-t001:** Different autophagic forms induced by microtubule poisons in different cancer types.

Agents	Cancer Type	Autophagy	Autophagy Inhibitor	Autophagy and Cell Death	Ref.
Colchicine	A549 lung cancer cell line	Autophagy induction (associated with senescence)	3-MA	Cytoprotective autophagy	[50]
Colchicine derivative “JG-03-14”	MCF-7 and MDA-MB-231 breast cancer cells	Autophagy induction	N/A	Cytotoxic autophagy	[52]
Colchicine derivative “JG-03-14”	B16/F10 melanoma and HCT-116 colon cancer cells	Autophagy induction	CQ and Baf A1	Cytotoxic autophagy	[53]
Colchicine derivative “Green 1”	PANC-1 pancreatic cancer and E6-1 or Jurkat acute T cell leukemia cell lines	Autophagy induction	N/A	Cytotoxic autophagy	[54]
Colchicine derivative “AD1”	U87MG and U373MG human malignant glioblastoma cell lines	Autophagy induction	N/A	Cytotoxic autophagy	[55]
Podophyllotoxin acetate	A549 and NCI-H1299 human non-small cell lung cancer cell lines	Autophagy induction	N/A	Cytotoxic autophagy	[60]
Podophyllotoxin derivative “Da-1”	K562/VCR chronic myeloid leukemia cell lines	Autophagy induction	N/A	Cytotoxic autophagy	[61]
Podophyllotoxin derivative “OAMDP”	HepG2 hepatoma cell line	Autophagy induction	N/A	Cytotoxic autophagy	[62]
Combretastatin A-4	MDA-MB-231 breast tumor cells, SGC-7901 human gastric tumor cells and SMMC-7721 human hepatocellular carcinoma cells	Autophagy induction	In vitro; 3-MA/Baf A1/siRNAs against Atg5 and Beclin 1 genes/BCL_2_ inhibitor (ABT-737) JNK inhibitor or JNK siRNA In vivo; 3-MA	Cytoprotective autophagy	[67]
Combretastatin A-4	CT-26 and HT-29 adenocarcinoma cell lines	Autophagy induction	3-MA, Baf A1	Non-protective autophagy in CT-26 cell line, Cytoprotective autophagy in HT-29 cell line	[68]
Combretastatin A-4	SJSA and MG63 human osteosarcoma cell lines	Autophagy induction	CQ	Cytoprotective autophagy	[69]
Combretastatin A-4 phosphate	PC3 prostate cancer xenografts	Autophagy induction	autophagy-defective PC3 prostate cancer xenografts (developed with retrovirally transducing PC-3 cells with ATG4BC74A)	Cytoprotective autophagy	[71,72]
Vinblastine	Ehrlich ascites tumor cells	Autophagy inhibition with autophagic vacuoles accumulation	N/A	N/A	[80]
Vinblastine	HepG2 human hepatocarcinoma and LS174T colon cancer cell lines	blocked autophagy maturation	CQ	Autophagy inhibition	[82]
Vincristine	Neuroblastoma cell lines	Autophagy induction	In vivo; *ATG5* knockdown In vivo; HCQ	Cytoprotective autophagy	[84]
Vincristine	Eca-109/VCR esophagus cancer cell line	Autophagy induction	autophagy inhibitor “4d”	Cytoprotective autophagy	[85]
Vincristine	REH and Nalm-6 pre-B acute lymphoblastic leukemia (ALL) cell lines	Autophagy induction	CQ	Cytoprotective autophagy	[86]
Vincristine	human retinoblastoma cells	Autophagy induction	CQ, CD24 knockdown	Cytoprotective autophagy	[87]
Vincristine	K562/ADM cell line	Autophagy induction	CQ reversed the inhibitory effect of MAT	MAT promote cytotoxic autophagy	[88]
Vincristine	multiple myeloma cells	Autophagy induction	N/A	Cotylenin A promote cytotoxic autophagy	[89]
Paclitaxel	breast cancer cells	Paclitaxel **prevents autophagosome maturation** and lysosome fusion in breast cancer cells	3-MA	Activates apoptosis as a result of autophagic flux inhibition in cancer cells	[98]
Paclitaxel	breast cancer cells	Paclitaxel alone **did not induce autophagy** in breast cancer cells, it enhanced ARHI-induced autophagy.	N/A	When ARHI was re-expressed in breast cancer cells treated with paclitaxel, the growth inhibitory effect of paclitaxel was enhanced in both the cell culture and the xenografts.	[97]
Paclitaxel	A2780, 3AO, and SKOV3 ovarian cancer cells	Paclitaxel increases autophagosome formation and autophagic flux	Beclin-1 deficiency	Cytoprotective autophagy	[99]
Paclitaxel	A549 cells; U87, PC3 and HT-29 cells	Autophagy induction (no p62/SQSTM1 detected)	3-MA /Beclin 1 siRNA	Cytoprotective autophagy	[100]
Docetaxel	LNCaP, PC3, and DU145	Autophagy induction	3-MA	Non-protective autophagy	[105]
Docetaxel	docetaxel resistant prostate cancer cell lines	Autophagy induction	N/A	Non-protective autophagy	[104]
Docetaxel	prostate cancer cells	Autophagy induction	3-MA	Cytoprotective autophagy	[106]
Epothilone A and Epothilone B	ovarian cancer cells	Autophagy induction	Baf A1	Cytoprotective autophagy	[113]
Ixabepilone	hepatic carcinoma, glioma cells and breast cancer cells	Autophagy induction	In vitro: CQ, 3-MA, beclin-1 si RNA In vivo: CQ	Cytoprotective autophagy	[116]
Ixabepilone	MDA-MB-231 and SUM159 cells	Ixabepilone increased p62/SQSTM1 expression, Ixabepilone either reduced or had no effect on the basal levels of LC3b-II	N/A	N/A	[115]

LC3: Microtubule-associated protein 1A/1B-light chain 3; CQ: chloroquine; HCQ: hydroxychloroquine; 3-MA: 3-Methyladenine; Baf A1: bafilomycin A1; OAMDP: 4β-(1,3,4-oxadiazole-2-amino-5-methyl)-4-deoxypodophyllotoxin; ATG4BC74A: an inactive and dominant-negative mutant of the autophagy related gene atg4B; MAT: Matrine; 4d: (N-(cyclohexylmethyl)-5-(((cyclohexylmethyl)amino) methyl)-2 ((4(trifluoromethyl)benzyl) oxy) benzamide); N/A: not applicable.

## Data Availability

Not applicable.

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
