# Peer review of "The Cytoprotective, Cytotoxic and Nonprotective Functional Forms of Autophagy Induced by Microtubule Poisons in Tumor Cells—Implications for Autophagy Modulation as a Therapeutic Strategy"

_biomedicines, 2022, doi:10.3390/biomedicines10071632_

Round 1
Reviewer 1 Report
Dear Sirs, I recommend accepting your manuscript in its present form. It is well-organized, very thoughtful, and novel on the topic of autophagy development and mechanism.
Author Response
Thank you for your positive comments relating to our manuscript. No concerns were raised that would have to be addressed.
Reviewer 2 Report
The manuscript gave a very comprehensive review of the role of autophagy in cancer treatment such as drug resistance induced by autophagy. The study was divided into different sections by different binding sites of different class drugs, which is a systematic method that is easy for readers to understand the relevant molecular mechanisms. The author highly recommends the publication of manuscript after authors consider make the following minor revisions:
1) Some sections content seemed to need rearrangements. For example, in section 3. Microtubule structure and functions, the third paragraph of this section seemed to be not closely relevant to the topic. Maybe this paragraph should be combined with the following sections that introduce the different drug mechanism sections.
2) Figure 1 can be further elucidated and noted. For example, what does it meaning by yellow arrow and green arrow? What is relationship of β-tub1 and β-tub2 with α-tubulin and β-tubulin. There should be a detailed figure legend in text but not just a figure caption.
3) Table 1 caption should be corrected because typo errors (different autophagic forms induced by microtubule poisons in different cancer types?). It is better for authors to give rationale to arrange the table in such a way because the information given in the table looks not well and systematic organized
Author Response
- Some sections content seemed to need rearrangements. For example, in section 3. Microtubule structure and functions, the third paragraph of this section seemed to be not closely relevant to the topic. Maybe this paragraph should be combined with the following sections that introduce the different drug mechanism sections.
Answer: Thank you for this suggestion. In section 3, our aim was to highlight the fact that the unique structure and function of microtubule are the primary reasons for their serving as targets for cancer therapeutic agents. Therefore, the third paragraph is actually required as a lead-in for the subsequent section on the relationship between microtubule poisons and autophagy. However, we recognize that the title of section 3 may be inexact and somewhat misleading; consequently, the subheading has been modified to read “3. Microtubules and microtubule poisons”.
- Figure 1 can be further elucidated and noted. For example, what does it meaning by yellow arrow and green arrow? What is relationship of β-tub1 and β-tub2 with α-tubulin and β-tubulin. There should be a detailed figure legend in text but not just a figure caption.
Answer: We have modified and improved the figure legend, as suggested. We added lines160-175 on page 4, highlighted in yellow.
- Table 1 caption should be corrected because typo errors (different autophagic forms induced by microtubule poisons in different cancer types?). It is better for authors to give rationale to arrange the table in such a way because the information given in the table looks not well and systematic organized
Answer: We have now corrected the table title to read “Different autophagic forms induced by microtubule poisons in different cancer types”. We also condensed some of the contents of table 1 to improve its organization.
Regarding the arrangement of the table, we organized the table according to the flow of the main text of the manuscript, to facilitate the readers’ ability to locate and link the papers cited in the table to the text contents.
The table also provides a brief summary of the text in terms of cancer types /the cell line used, whether autophagy is induced or inhibited in response to the microtubule poisons used, the experimental approaches utilized for autophagy inhibition and the conclusion of each paper in terms of the type of autophagy induced.
Reviewer 3 Report
In this article, Jingwen Xu et al. comprehensively summarized the literature surrounding the microtubule poisons and their bewildering effects on autophagy in cancer cells. The authors should be commended for their efforts in extensively summarizing the vast literature in this area. The article serves as a guide for all oncology researchers to get a preliminary idea on how complex effects on autophagy can be induced by microtubule inhibitors. I think the article is appropriate for the readers of Biomedicines and should be considered for possible publication, provided the authors address the following concerns.
1. The fact that the autophagy induced by different classes of drugs despite having a common binding site on microtubules (for instance OAMDP and Combrestatin A4 have same binding site yet they induced cytotoxic and cytoprotective forms of autophagy in hepatic cancer cells) indicates the fact that microtubule poisons themselves have different influence on autophagy possibly because of their chemical structure differences? Authors may consider adding few lines in the discussion supporting/opposing this claim.
2. Autophagy is a complex process and has yin yang role in the cellular homeostasis. This is the possible reason that studies concerned with autophagy report contrasting outcomes. Often this is ascribed to the difference in the role played by the autophagy in acute vs chronic stages of the disease. Do authors think, if induction of cytoprotective or cytotoxic forms of autophagy also depends on the stage of the disease (especially for in vivo models) and duration of the treatment (for in vitro studies)? If there is any such information available in the literature, it is encouraged to be discussed in this manuscript that ultimately benefit the readership.
3. The title of the manuscript mentioning autophagy inhibition as a therapeutic strategy reflects a biased version of looking at autophagy. If the autophagy induced by microtubule poisons is cytotoxic then obviously autophagy induction rather than inhibition is a beneficial strategy to augment the cytotoxic effects. In account of this contradiction, I think it’s better to mention it as “implications for autophagy modulation as a therapeutic strategy” in the title.
4. Page number 6, line number 252, JC-03-14 should be replaced with JG-03-14.
Author Response
- The fact that the autophagy induced by different classes of drugs despite having a common binding site on microtubules (for instance OAMDP and Combrestatin A4 have same binding site yet they induced cytotoxic and cytoprotective forms of autophagy in hepatic cancer cells) indicates the fact that microtubule poisons themselves have different influence on autophagy possibly because of their chemical structure differences? Authors may consider adding few lines in the discussion supporting/opposing this claim.
Answer: Thank you for your guidance. This is a valid point, although the literature does not currently provide any direct guidance. We have added the following sentence to the discussion (page 14) “This may be dependent upon the chemical structure of the drug as well as the experimental cell line being utilized” .
- Autophagy is a complex process and has yin yang role in the cellular homeostasis. This is the possible reason that studies concerned with autophagy report contrasting outcomes. Often this is ascribed to the difference in the role played by the autophagy in acute vs chronic stages of the disease. Do authors think, if induction of cytoprotective or cytotoxic forms of autophagy also depends on the stage of the disease (especially for in vivo models) and duration of the treatment (for in vitro studies)? If there is any such information available in the literature, it is encouraged to be discussed in this manuscript that ultimately benefit the readership.
Answer: Thank you for this highly intriguing query. As the reviewer indicates, one of the fundamental challenges in this area is to identify the basis for autophagy exhibiting different roles throughout the in vitro experimental literature. However, as with many elements in this field, we cannot extract this type of information from the data available utilizing in vitro cancer cell models. For example, whereas we have treated tumor cells with cisplatin in the presence of multiple autophagy inhibitors at different time points, the role of autophagy was not dependent on the treatment duration, but was related only to the cell genotype (such as p53 status). It is possible that future studies in our laboratory (or that of others) will provide insights into this question.
- The title of the manuscript mentioning autophagy inhibition as a therapeutic strategy reflects a biased version of looking at autophagy. If the autophagy induced by microtubule poisons is cytotoxic then obviously autophagy induction rather than inhibition is a beneficial strategy to augment the cytotoxic effects. In account of this contradiction, I think it’s better to mention it as “implications for autophagy modulation as a therapeutic strategy” in the title.
Answer: We appreciate this suggestion and we have modified the title, as suggested.
- Page number 6, line number 252, JC-03-14 should be replaced with JG-03-14.
Answer: Thank you for noting this error. JC-03-14 has been corrected to read JG-03-14.